# TCRN: A Two-Step Underwater Image Enhancement Network Based on Triple-Color Space Feature Reconstruction

**Sen Lin** , **Ruihang Zhang** *, **Zemeng Ning and Jie Luo**

School of Automation and Electrical Engineering, Shenyang Ligong University, Shenyang 110159, China; linsen@sylu.edu.cn (S.L.); ningzemeng98@gmail.com (Z.N.); luojie0013@gmail.com (J.L.)
* Correspondence: zhangruihang@sylu.edu.cn

**Abstract:** The underwater images acquired by marine detectors inevitably suffer from quality degradation due to color distortion and the haze effect. Traditional methods are ineffective in removing haze, resulting in the residual haze being intensified during color correction and contrast enhancement operations. Recently, deep-learning-based approaches have achieved greatly improved performance. However, most existing networks focus on the characteristics of the RGB color space, while ignoring factors such as saturation and hue, which are more important to the human visual system. Considering the above research, we propose a two-step triple-color space feature fusion and reconstruction network (TCRN) for underwater image enhancement. Briefly, in the first step, we extract LAB, HSV, and RGB feature maps of the image via a parallel U-net-like network and introduce a dense pixel attention module (DPM) to filter the haze noise of the feature maps. In the second step, we first propose the utilization of fully connected layers to enhance the long-term dependence between high-dimensional features of different color spaces; then, a group structure is used to reconstruct specific spacial features. When applied to the UFO dataset, our method improved PSNR by 0.21% and SSIM by 0.1%, compared with the second-best method. Numerous experiments have shown that our TCRN brings competitive results compared with state-of-the-art methods in both qualitative and quantitative analyses.

**Keywords:** underwater image enhancement; visual color correction; feature reconstruction; deep learning

## 1. Introduction

Underwater image processing is a fundamental vision task of marine autonomous detectors that can greatly advance ocean research and underwater engineering and is used for underwater scene analysis [1], fish recognition, 3D reconstruction [2], etc. However, underwater images always suffer from a deterioration in quality. On the one hand, a blue or green cast can occur due to the different attenuation rates of the various wavelengths of light propagating through the water. On the other hand, the light can be scattered by floating particles and other impurities present in the water body, which ultimately limits the contrast of the image [3]. These limitations have a significant negative impact on the acquisition of high-quality underwater images.

To address these limitations, previous researchers have used traditional methods of underwater image processing, which can be broadly categorized as image enhancement and physical model-based image restoration. Image enhancement methods such as histogram stretching [4], gray world [5], and white balance [6] either try to directly change the pixel value for enhancement, without considering the underwater imaging process, or convert the image to the frequency domain using the wavelet transform and Fourier transform methods to improve the image indirectly by adjusting the frequency domain characteristics. However, the noise caused by the haze effect in the frequency domain cannot be effectively eliminated by these methods. Physical model-based image restoration methods can be used to build the imaging model [7] and employ some priors [8,9] to estimate the model

parameters to invert the real image. However, priors are not always accurate in this context because of the complexity of underwater situations.

Over the past few years, deep-learning-based methods [10–20] for underwater image enhancement have grown rapidly with the widespread use of artificial intelligence in a variety of fields. Simply put, deep learning is a method that adapts to the non-linear relationship between the real images and the degraded images by training using a dataset. Depending on the types of networks, deep-learning approaches for enhancing underwater images can be roughly divided into generative adversarial networks (GANs) [10–15] and convolutional neural networks (CNNs) [16–20]. However, most networks either focus only on RGB feature processing or do not make full use of other color space information, so the accuracy and visual quality of the result with enhanced color cannot be guaranteed. There are several proposed methods [21,22] for creating a color space loss function that can force the network to learn more features of the different color spaces or to utilize semi-supervised [23] or unsupervised [24] learning strategies, in order to address the negative limitations associated with underwater images. However, the haze noise is not processed separately in these networks, which may cause the residual interference information to be enhanced along with the critical areas in subsequent processes.

From a training perspective, using triple-color spaces as inputs allows the network to learn more of the features of a single image. The technique differs from methods using image rotation as a means of enlarging the training set since our operation did not increase the number of samples in the training set. Although these color spaces can be mathematically converted, they represent distinct physical quantities that are crucial for image enhancement. Compared with a single RGB input, triple-color spaces give the network a stronger fitting ability. Therefore, we designed a feature extraction module for RGB, LAB, and HSV spaces as the first step of the network, aiming to solve color distortion in triple-color space, and introduced a dense pixel attention module (DPM) based on dense connection to filter features. Then, we carried out feature fusion and reconstruction in the second step of the network. As Figure 1 indicates, compared to the raw images, our method successfully restored the image quality and predicted a more consistent result when working with human visual perception. The main contributions of this paper are as follows.

1.　We designed an end-to-end triple-color space feature extraction network coupled with DPM to achieve cleaner visual characteristic information.
2.　We reinforced the long-term dependency of the different color spaces from a fully connected layer perspective and utilized a group structure for spatial reconstruction.

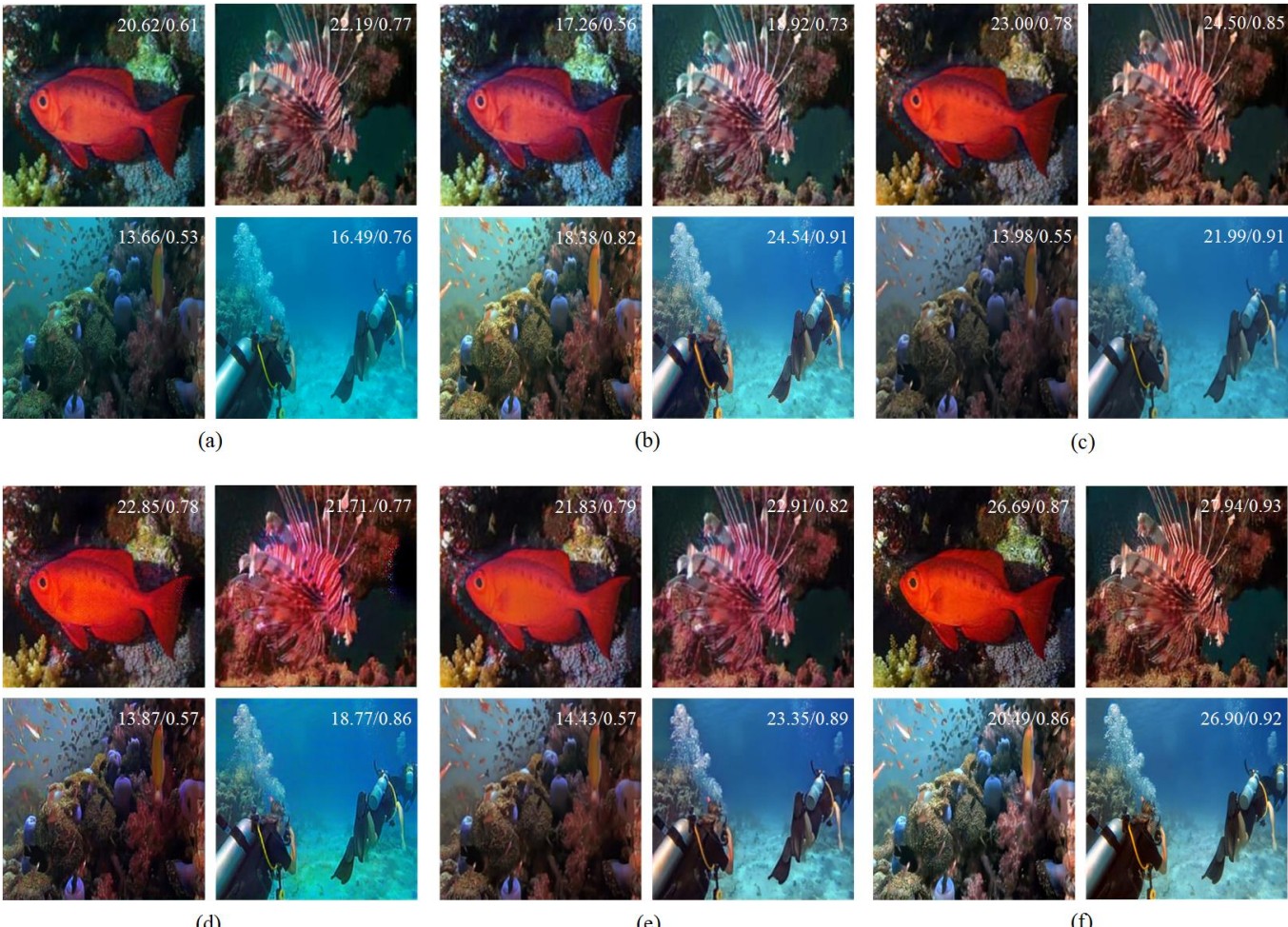

**Figure 1.** Intuitive comparisons of the different methods for enhanced images. (**a**) Raw images. (**b**) Water-net. (**c**) Shallow-uwnet. (**d**) FUnIE-GAN. (**e**) Uiess. (**f**) The model presented herein. Our model removes the color cast and haze effectively. Compared with the existing methods, this novel TCRN achieves the best PSNR and SSIM scores.

## 2. Related Works

Deep-learning-based enhancement methods are classified by researchers according to different perspectives. In our work, we roughly distinguish these methods as CNN-based methods and GAN-based methods, which we will discuss in detail in this section.

GAN-based methods: Gans utilize generators and discriminators against each other to update the weight parameters and, finally, acquire an enhanced image. Li et al. [10] constructed a two-stage underwater generative adversarial model. In the first stage of the model, the style migration characteristic of Gan was utilized to produce a synthetic dataset to train the second stage of the model, then unsupervised color correction was carried out on the monocular underwater images. Guo et al. [11] designed a multi-scale densely connected module and added it to the generators to improve network performance and render the images in more detail. Yang et al. [12] transformed the discriminator into a dual discriminator, forcing the generator to focus on the global and local semantic information. However, the color correction of the processed image was incomplete. Li et al. [13] proposed a multi-input Gan and combined different loss functions to solve the color deviation of degraded images; however, the restored images still show the haze effect. Gans combined with physical models have also been proposed [14,15]. However, the restored images did not achieve a realistic visual result.

CNN-based methods: The CNN's powerful feature extraction capability enables deep learning networks to learn image details and access global semantic information. In one study [16], a sequential residual channel attention module was proposed to extract crucial features, then, at the end of the network process, the image was converted to a YUV color space for brightness enhancement. Qi et al. [17] were the first to introduce collaborative processing and a joint learning strategy to achieve underwater image enhancement. They designed a semantic correlation feature-matching algorithm to interrelate two branches of an encoder-decoder network to realize complementary learning. Inspired by the preceding study, Li et al. [18] proposed that lightweight networks could be used to directly enhance degraded images, which greatly improved the computing speed. Xiao et al. [19] designed a dual-statistical white balance algorithm based on deep learning, which simultaneously stretched the RGB, LAB, and HSI to achieve color correction and brightness improvement. Li et al. [20] combined the traditional method with an encoder-decoder network, using the encoder-decoder network to extract multicolor space features and compensate for the features by calculating inverse depth maps. Although the enhanced image has good results in terms of color correction, the utilization of multicolor spaces is too nominal in the encoder stage.

These deep-learning-based methods either improve image quality by extracting features from a single color space or fail to fully utilize the feature information from other color spaces. As a result, the recovered image always presents a single visual style. In contrast, our method uses triple-color space features to reconstruct the vivid color style of the image and add the DPM to ensure the elimination of the haze effect.

## 3. Proposed Method

We present the architecture of TCRN in Figure 2. The model includes two steps: triple-color space feature extraction and image reconstruction based on feature fusion. In STEP I, the RGB input is first converted into LAB and HSV spaces, then three images are passed through parallel U-net-like paths, known as the triple-color space feature extraction network, to collect different visual characteristics, which are named $f_{rgb}$, $f_{hsv}$, and $f_{lab}$; $c$ represents the number of channels. In each long skip concatenation, the DPM and the pre-fusion module are also added to help each path to obtain clean feature information regarding the other color spaces. Finally, the features of all the color spaces are concatenated in the channel dimension. In STEP II, we use global average pooling to obtain the most representative number of $c$ neurons of the concatenated features; these neurons create a fully connected layer to decide every new neuron together, to compensate for the concatenated features. There are two reasons for this operation. The first is to strengthen the dependency between the features of the different color spaces; this prevents feature separation due to the depth of every single path [25]. The second is that we use the fully connected layer to learn the correlation losslessly, instead of using a common bottleneck connection. At the end of the network process, the group structure completes the image reconstruction.

In the following subsections, we describe all the components of our method in detail, including the feature extraction module (Section 3.1), the pre-fusion module (Section 3.2), the dense pixel attention module (Section 3.3), the group structure (Section 3.4), and the loss function (Section 3.5).

### 3.1. Feature Extraction Module

Figure 3a shows the details of the feature extraction module. Notably, some components of these color spaces may contain similar information, but in each U-net-like branch, the features of one component cannot be extracted accurately without the support of other components. Therefore, we kept the full RGB, LAB, and HSV features as inputs. To better extract the different types of visual characteristic information, we designed a sequential residual structure to prevent the loss of details and reduce difficulty in network training. Each feature extraction module is followed by one 2× global max pooling, the input channels are doubled from 64 after each module (64, 128, 256, and 512).

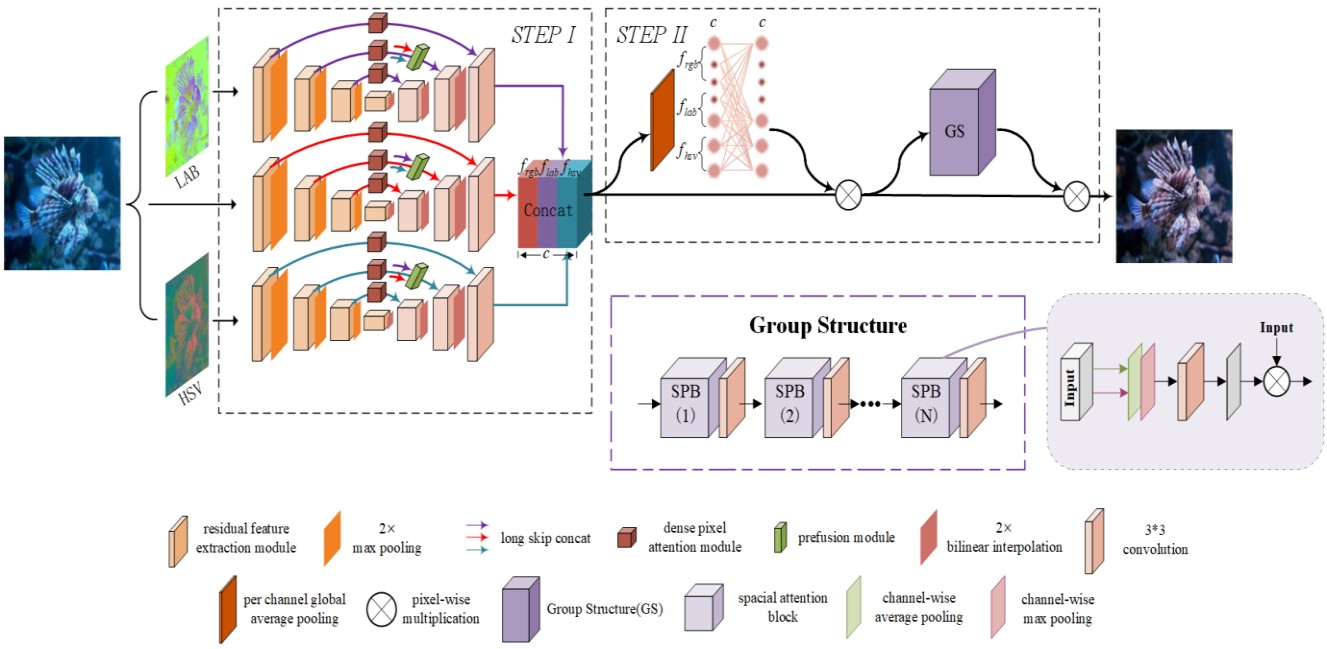

**Figure 2.** The framework of the TCRN. Our TCRN consists of triple-color feature extraction (STEP I) and construction based on feature fusion (STEP II). $f_{rgb}$, $f_{hsv}$, and $f_{lab}$ denote the features of RGB, HSV, and LAB, respectively. $c$ represents the number of channels.

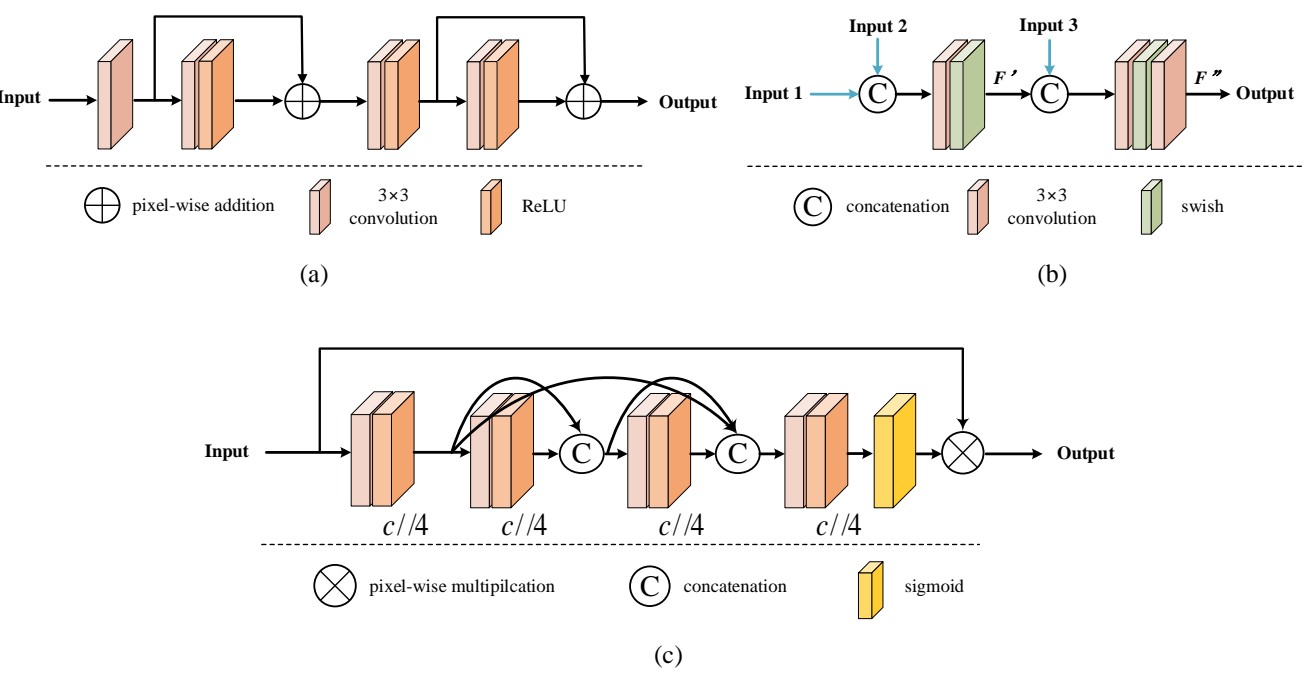

**Figure 3.** A schematic illustration of the proposed models. (**a**) Feature extraction module. (**b**) Prefusion module. (**c**) Dense pixel attention module. $c$ represents the channels of the input.

### 3.2. Dense Pixel Attention Module

Instead of the u-net's direct traditional skip connection, we added pixel-level attention to focus on feature value before skip concatenation. Generally, pixel attention processes the features unequally by adjusting the weight parameters to strengthen crucial information and filter out noise. Inspired by an earlier study [26], we first employed a more aggressive pixel attention strategy, named DPM, to differentiate the quality of information by taking advantage of the fact that dense connections encourage feature reuse. As shown in Figure 3c,

first, a convolution and ReLU are used to capture the input information, whereby the shape changes from c × H × W to c//4 × H × W. Then, the visual feature information is strengthened by a dense connection composed of two convolutions and ReLU, while the shape of the features remains c//4 × H × W. Finally, a convolution and ReLU are used to change the shape into 1 × H × W, and the sigmoid function maps the pixel values as weights, multiplied by the input.

From a systemic perspective, compared to other, similar, U-net-like networks, the DPM on the skip concatenation prevents information loss so that the initial feature is successfully transmitted to the deeper network. This operation implicitly enables deep supervision. Meanwhile, due to the structure of the dense connection, backpropagation can be achieved more easily.

### 3.3. Pre-Fusion Module

In order to consider all the features of the different color spaces in the same pixel position without encountering the feature separation phenomenon, a pre-fusion module is proposed, as shown in Figure 3b. To optimize the module's nonlinear fitting ability for different visual characteristics, the swish function [27] is used in the activation layer, which can be expressed thus:

$$swish(x) = x \times \frac{1}{1 + e^{-\beta x}} \tag{1}$$

where $\beta \in [0, \infty)$ is a trainable parameter in our work, $\beta = 1$.

Assuming that the input is $\in F^{S \times H \times W}$, where $F$ is a feature map from one color space, $S \in (RGB, LAB, HSV)$, and $H$ and $W$ are the height and width of the input, respectively. Inputs 1, 2, and 3 come from the different color spaces and are named $I_1$, $I_2$, and $I_3$, and have been marked with colors in Figure 2. Thus, $F'$ and $F''$ can be expressed thus:

$$F' = s(conv(cat(I_1, I_2))) \tag{2}$$

$$F'' = conv(s(conv(cat(F', I_3)))) \tag{3}$$

where $conv(\cdot)$ indicates the convolution operation, $s(\cdot)$ represents the swish function, and $cat(\cdot)$ indicates the feature concatenation.

### 3.4. Group Structure

As shown in Figure 2, we designed a spatial pixel reconstruction group structure to better integrate the features and reconstruct the images. The group structure is composed of 8 spacial attention blocks (SPBs). Given that the features in the different color spaces make diverse contributions to the reconstructed images, in each SPB, we adopt channel-wise maximum pooling and average pooling, respectively, to obtain two spatial sample layers, then use a 3 × 3 convolution to form a weight layer that is multiplied by the input. The ablation study in Section 4.6 shows the selection of the number of SPBs in more detail.

### 3.5. Loss Function

To preserve the data fidelity, we selected a linear combination of $L_1$ loss and VGG perceptual loss to train our network.

The $L_1$ loss measures the absolute distance between the truth result, $I_g$, and the reconstructed result, $I_P$, to explicitly fit the network. It can be represented by:

$$L_1 = \sum_{m=1}^{H} \sum_{n=1}^{W} \left| I_g(m, n) - I_p(m, n) \right|. \tag{4}$$

Our VGG perceptual loss is based on a VGG-16 pre-trained model. We used the last convolution layer of VGG-16, defined as $\varphi(\cdot)$, to calculate the feature similarity between $I_g$ and $I_P$ to implicitly train the network. It can be expressed as:

$$L_{VGG} = \sum_{m=1}^{H} \sum_{n=1}^{W} \left| \varphi(I_g)(m,n) - \varphi(I_P)(m,n) \right|. \tag{5}$$

Then, the total loss can be defined as:

$$L_f = L_1 + \lambda L_{VGG} \tag{6}$$

where $\lambda$ is constantly set to 0.5 for the purposes of balancing the values of different losses.

## 4. Experiments

In this section, first, the datasets used for training and the evaluation metrics are described. Then, we present details related to the implementation of the training. We qualitatively and quantitatively compare the process with other methods on different datasets; these methods include GDCP [28], UDCP [9], Fusion [29], UIEIVM [30], Uiess [31], Water-net [32], Shallow-uwnet [33], and Funie-Gan [34]. Finally, a sufficient number of ablation experiments are conducted to verify the contributions of the proposed modules.

### 4.1. Benchmarks

As shown in Table 1, we used five benchmarks independently in our experimental environment; these are Dark [34], Uimagine [34], UIEB [32], UFO [35], and Color-Check7 [29]. These datasets of different scales contain a variety of underwater situations, which can be used as a challenging test for our method.

**Table 1.** The presentation of dataset divisions. The numbers represent the number of images in each training set and test set.

| | Train Set | Test Set | |
| | | Paired | Unpaired |
|---|---|---|---|
| UIEB | Train-800 | Test-90 | Test-60 |
| Uimagine | Train-3700 | Test-515 | None |
| UFO | Train-1500 | Test-120 | None |
| Dark | Train-5500 | None | Test-570 |

Uimagine contains 3700 pairs of training images that are used to form the training set, named Train-3700, and 515 pairs of images that are used to form the test set, named Test-515. The Dark dataset includes a large number of dark deep-sea scenes. From this dataset, 5500 pairs of images were used and named Train-5500, and 570 unpaired pictures were used and named Test-570. In the UIEB dataset, 800 pairs of images in the training set were randomly selected for training and named Train-800, while the remaining 90 pairs of images were named Test-90. In total, 60 challenging images were used to test the robustness of our network in this dataset, which was named Test-60. In the UFO dataset, 1500 pairs of images were used as the Train-1500 set and 120 pairs of images were used as the Test-120 set. Color-Check7 includes seven underwater color cards with images taken by different cameras and a reference color card. These cards were taken using a CanonD10, FujiZ33, OlympusT6000, OlympusT8000, PentaxW60, PentaxW80, and PanasonicTS1, which were named D10, Z33, T6000, T8000, W60, W80, and TS1, respectively.

### 4.2. Evaluation Metrics

For our test sets in Uimagine, Dark, UIEB, and UFO, we used three full-reference evaluation metrics and three no-reference evaluation metrics to estimate the enhancement quality. Briefly, full-reference evaluation metrics were used, including peak signal-to-

noise ratio (PSNR) [36], the patch-based contrast quality index (PCQI) [37], and structural similarity (SSIM) [38]. PCQI calculates the signal intensity and signal structure between the label image and the predicted image in each patch, whereby a higher PCQI means higher contrast of the predicted image. PSNR represents the information content gap between the predicted image and the label image, whereby a higher PSNR means a better effect in terms of details, while a higher SSIM represents more similarity in terms of structural content. These full-reference evaluation metrics are used for the paired Test-90, Test-120, and Test-515 (paired test sets). The no-reference evaluation metrics include UIQM [39], UCIQE [40], and EMBM [41]. UIQM addresses color deviation, contrast, and brightness, which can reflect visual performance. UCIQE evaluates the saturation of underwater images and EMBM depicts the pixel contrast at the edges of objects. Higher scores for these no-reference metrics, which were used on Test-60 and Test-570 (paired test sets), indicate better-enhanced performance. However, it is worth mentioning that the no-reference index is not an accurate representation of true image quality; the scores of no-reference evaluation metrics are given as a reference in our research.

In the Color-Checker7 evaluation, we used CIEDE2000 to score all the predicted results. Basically, the color panel consists of 24 color patches; we crop the color patches predicted by all the methods and compare these patches with the reference color card to calculate the color deviation.

### 4.3. Implementation Details

We chose Python 3.7 as the programming language on Ubuntu18.04 and used the Py-Torch framework with the Adam optimizer to train our network via the NVIDIA RTXA5000. The initial learning rate was set as 0.01. Then, 0.9 and 0.999 were chosen as the default values for parameters $\beta 1$ and $\beta 2$, respectively. We used a cosine annealing strategy [42] to gradually transform the learning rate from the initial value to 0. Assuming that the total batch number is set to $B$ and the initial learning rate is $\delta$, then, at batch $b$, the learning rate $\delta_b$ is expressed as:

$$\delta_b = \frac{1}{2}(1 + \cos(\frac{b\pi}{B}))\delta. \tag{7}$$

The input size was resized to $256 \times 256$. The caveat is that the image sizes are not always consistent with the requirements for UIEB and UFO; the images under the two datasets were adjusted to $256 \times 256$ since our current task does not concern high resolution.

### 4.4. Quantitative Comparisons

Quantitative comparisons were conducted on the Test-60, Test-120, and Test-515 datasets. The PCQI, SSIM, and PSNR scores of different methods are shown in Table 2. As can be seen, our TCRN achieved competitive results on Test-90, Test-120, and Test-515. In addition, there are three interesting findings from the quantitative analysis: (1) traditional methods generally score low in the evaluation, which reflects the finding that image enhancement or restoration based on traditional methods often improves the performance of the raw image at the expense of the original structure and color fidelity. (2) In the PCQI scores, all the methods cannot exceed the original image. During the experiment, we found that some predicted images achieved extremely low PCQI scores, which led to a decrease in the average value. (3) Our method outperformed on the small datasets of UIEB and UFO, which means that our TCRN can learn better mapping with a small number of samples.

Next, we employed EMBM, UIQM, and UCIQE to evaluate the unpaired datasets of Test-60 and Test-120. The performance values of all methods is shown in Table 3. From Table 3, it can be seen that our TCRN achieves the second-best UIQM and UCIQE scores on Test-60, and the best EMBM scores on Test-570. It should be noted that the no-reference metrics scores cannot fully reflect the actual image quality. More color distortion may lead to different UIQM/UCIQE scores, while over-sharpening the edges of objects can increase the EMBM scores; these conditions result in failure enhancement cases.

**Table 2.** Underwater image enhancement performance metrics in terms of the average PCQI, PSNR, and SSIM values. We represent the two best results in red and blue colors.

| Models | Test-90 (UIEB) | | | Test-120 (UFO) | | | Test-515 (Uimage) | | |
|---|---|---|---|---|---|---|---|---|---|
| | PCQI | PSNR | SSIM | PCQI | PSNR | SSIM | PCQI | PSNR | SSIM |
| raw | 1.126 | 16.40 | 0.749 | 0.819 | 20.32 | 0.760 | 0.894 | 20.03 | 0.791 |
| UDCP | 0.942 | 12.99 | 0.608 | 0.678 | 16.82 | 0.635 | 0.732 | 16.36 | 0.634 |
| Fusion | 0.979 | 20.59 | 0.873 | 0.648 | 18.26 | 0.733 | 0.691 | 18.76 | 0.775 |
| GDCP | 0.821 | 14.18 | 0.732 | 0.496 | 14.02 | 0.634 | 0.517 | 13.33 | 0.647 |
| UIEIVM | 0.741 | 20.38 | 0.839 | 0.430 | 17.06 | 0.663 | 0.456 | 17.15 | 0.683 |
| Water-net | 0.982 | 21.13 | 0.858 | 0.661 | 19.40 | 0.749 | 0.704 | 19.70 | 0.790 |
| Shallow-uwnet | 1.060 | 16.11 | 0.727 | 0.755 | 22.16 | 0.760 | 0.800 | 22.58 | 0.793 |
| FUnIE-GAN | 0.885 | 16.72 | 0.726 | 0.662 | 21.39 | 0.741 | 0.702 | 23.41 | 0.801 |
| Uiess | 0.926 | 18.48 | 0.781 | 0.705 | 20.46 | 0.758 | 0.741 | 20.74 | 0.793 |
| TCRN | 0.941 | 22.13 | 0.905 | 0.776 | 26.75 | 0.835 | 0.778 | 22.59 | 0.794 |

**Table 3.** Underwater image enhancement performance metric in terms of the average UIQM, UCIQE, and EMBM values. We represent the two best results in red and blue colors.

| Models | Test-60(UIEB) | | | Test-570 (Dark) | | |
|---|---|---|---|---|---|---|
| | UIQM | UCIQE | EMBM | UIQM | UCIQE | EMBM |
| raw | 0.383 | 0.366 | 0.563 | 1.116 | 0.419 | 0.486 |
| UDCP | 0.313 | 0.504 | 0.531 | 1.501 | 0.486 | 0.482 |
| Fusion | 0.622 | 0.427 | 0.523 | 1.024 | 0.404 | 0.482 |
| GDCP | 0.616 | 0.445 | 0.498 | 1.610 | 0.465 | 0.485 |
| UIEIVM | 1.176 | 0.479 | 0.461 | 2.178 | 0.472 | 0.468 |
| Water-net | 0.605 | 0.432 | 0.498 | 1.067 | 0.423 | 0.476 |
| Shallow-uwnet | 0.366 | 0.341 | 0.481 | 0.919 | 0.414 | 0.442 |
| FUnIE-GAN | 0.521 | 0.415 | 0.556 | 1.017 | 0.402 | 0.471 |
| Uiess | 0.513 | 0.418 | 0.460 | 0.995 | 0.414 | 0.442 |
| TCRN | 0.655 | 0.481 | 0.476 | 0.845 | 0.361 | 0.487 |

For ColorChecker 7, the CIEDE2000 scores are shown in Table 4 to verify the robustness and measure the color bias across all methods. TCRN demonstrates excellent performance with the D10, TS1, W60, and W80. Furthermore, our method achieves the highest average score. The result proves that our model restores the color to the greatest extent when using different cameras.

**Table 4.** The color dissimilarity comparisons of the different methods, using ColorChecker 7 in terms of the CIEDE2000 scores. We represent the two best results in red and blue colors.

| Models | D10 | Z23 | T6000 | T8000 | W60 | W80 | TS1 | Avg |
|---|---|---|---|---|---|---|---|---|
| raw | 22.19 | 23.26 | 23.28 | 29.08 | 22.07 | 26.17 | 24.34 | 24.34 |
| UDCP | 24.41 | 25.08 | 22.40 | 30.89 | 21.17 | 27.93 | 24.64 | 25.22 |
| Fusion | 19.63 | 18.95 | 16.46 | 19.52 | 20.19 | 24.54 | 19.34 | 19.80 |
| GDCP | 25.47 | 24.12 | 22.78 | 22.90 | 23.04 | 30.85 | 21.14 | 24.33 |
| UIEIVM | 22.64 | 21.93 | 19.75 | 22.83 | 21.86 | 23.40 | 19.35 | 21.68 |
| Water-net | 19.41 | 19.80 | 17.74 | 24.60 | 19.35 | 21.00 | 19.24 | 20.16 |
| Shallow-uwnet | 20.25 | 18.86 | 17.15 | 27.45 | 19.78 | 25.06 | 27.62 | 22.31 |
| FUnIE-GAN | 23.99 | 21.21 | 18.42 | 27.02 | 21.20 | 24.53 | 28.29 | 23.52 |
| Uiess | 19.42 | 21.31 | 17.69 | 23.54 | 17.67 | 19.63 | 21.46 | 21.10 |
| TCRN | 19.12 | 19.00 | 18.90 | 21.50 | 17.45 | 18.04 | 16.13 | 18.59 |

*4.5. Qualitative Comparisons*

In this section, we discuss several visual comparisons with other contrast methods.

First, we show all the predicted ColorChecker images. We take the examples of an underwater color card captured by a Panasonic TS1, shown in Figure 4. As can be seen, the raw image suffers from a severe pinkish cast. Most methods are not able to achieve a satisfactory result. The GDCP over-enhances the brightness at the cost of destroying the structure of the color patches, which leads to a poor score, while our TCRN restores the color and achieves the best score in terms of the CIEDE2000.

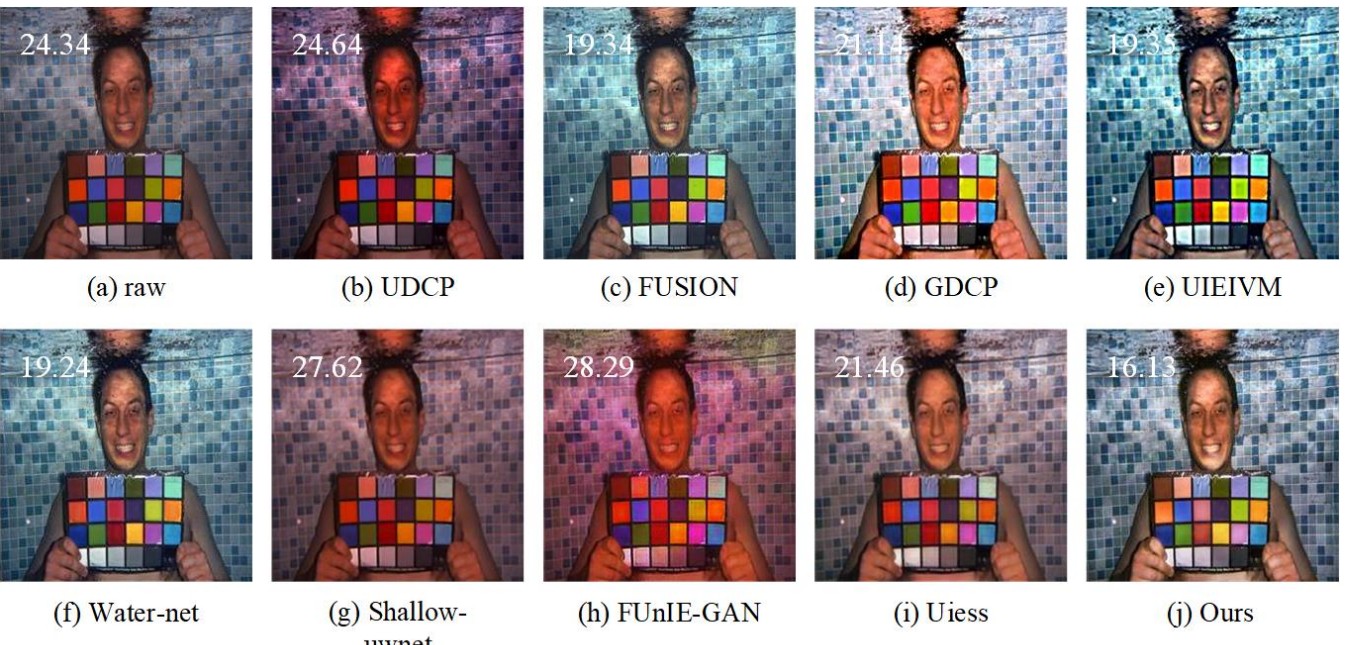

**Figure 4.** Visual comparisons of an underwater image taken by a Panasonic TS1 camera.

We then intuitively show the effect of color reconstruction in Figure 5. In order to avoid any preconceived influence on subjective visual perception, all images in Figure 5 were not labeled with metrics. We chose four images using the Test-60 and Test-120 datasets that suffered from severe yellow, blue, green, and dark color deviations, respectively. The gray world theory [43] considers that the R, G, and B components of a regular color image tend to be in balance. The color correction results of the compared methods can be visually revealed in the scatter distribution of RGB space. (Note that, as shown in Table 2, the full reference scores for the traditional methods are generally lower than deep-learning-based models. Owing to length limitations, only the images enhanced by deep learning are shown in Figures 5 and 6). In the yellow deviation image, other models failed to reconstruct a proper color distribution, while Water-net even produces a color distortion. Our TCRN effectively stretches the color of the pixel to ensure visual perception. In blue and green color deviation images, although Uiess can improve the image quality to a certain extent, TCRN achieves a more convergent distribution in the RGB color space. In challenging dark images, the stretching of R, G, and B components by our model achieves balance in the whole color space. TCRN restores not only the brightness of the scene but also the Tyndall effect.

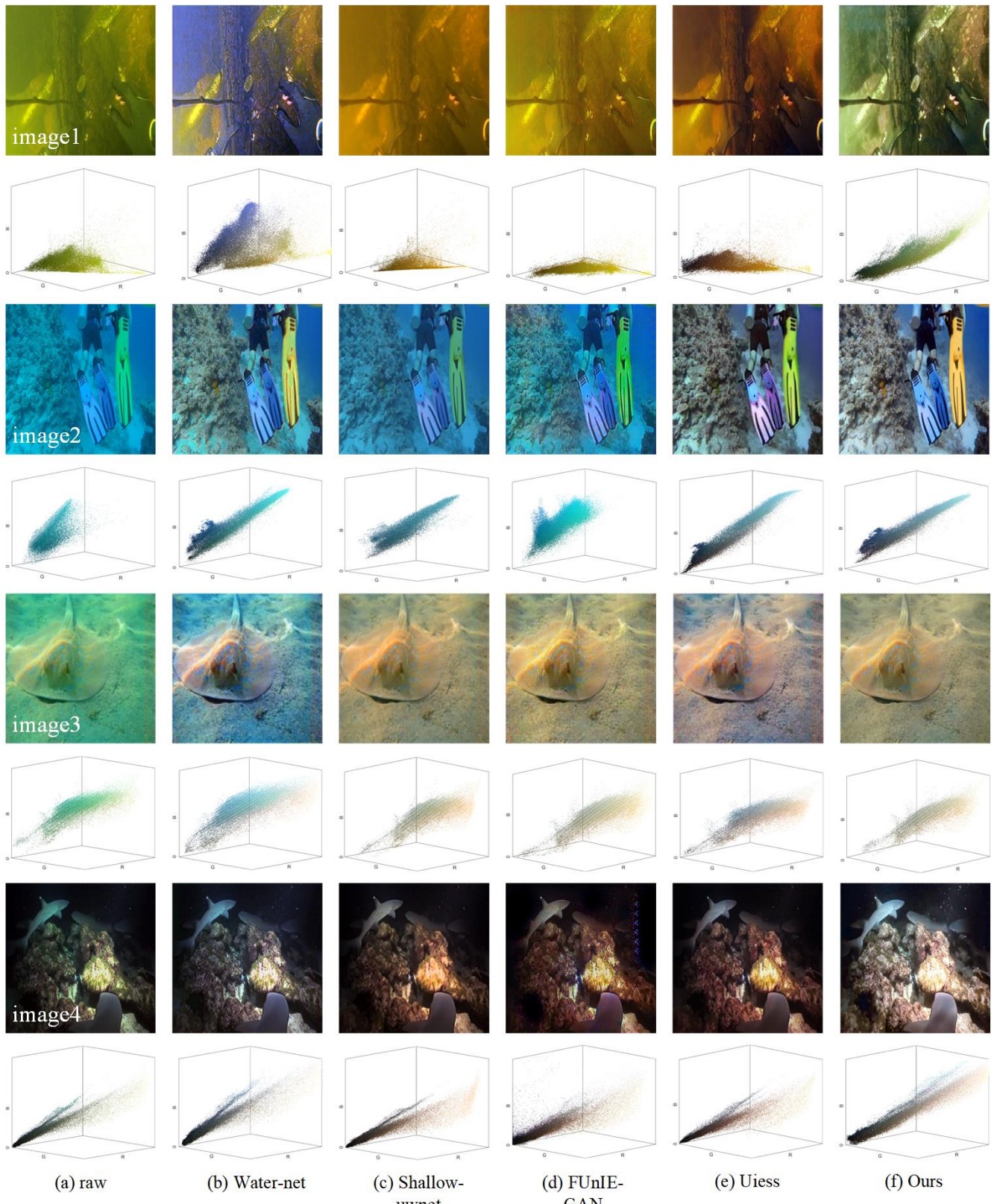

**Figure 5.** Color balance comparison results on the Test-60 and Test-120 datasets.

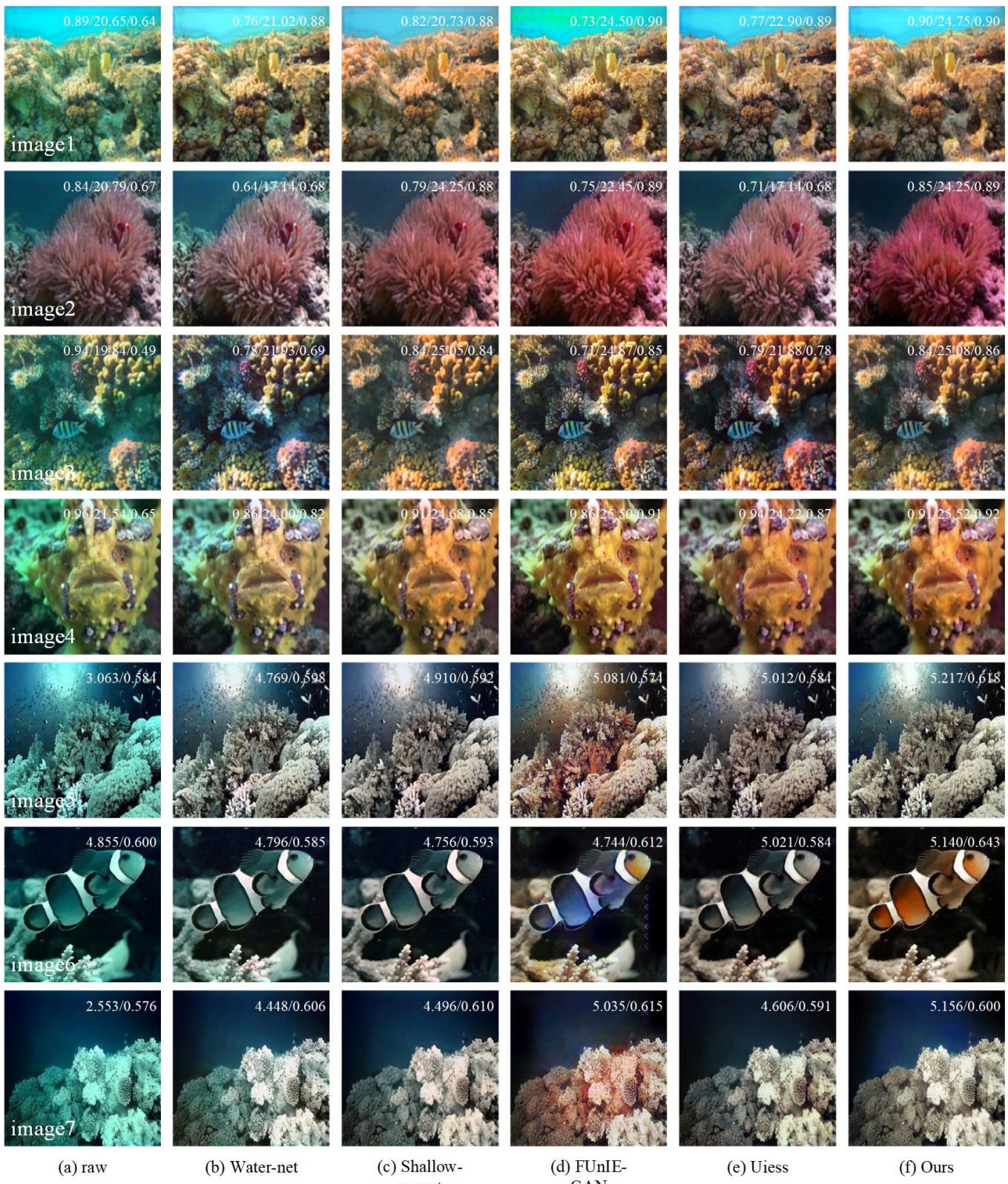

|         |           |           |           |          |          |
| ------- | --------- | --------- | --------- | -------- | -------- |
| (a) raw | (b) Water-net | (c) Shallow-uwnet | (d) FUnIE-GAN | (e) Uiess | (f) Ours |

**Figure 6.** Visual comparisons of images sampled from the Test-515 and Test-570 datasets.

To verify the ability of TCRN to improve contrast and restore visual perception in different underwater environments, we selected seven images with different degradation conditions in the paired dataset of Test-515 and the unpaired dataset of Test-570 for visualization experiments. These are presented in Figure 6, including images of underwater low

contrast, underwater turbidity, underwater complex structures, and underwater low-light scene. The numbers on the first four images refer to the PCQI/PSNR/SSIM. And the numbers on the last three images refer to the UIQM/UCIQE. In image1 and image2, our model accurately removes the color deviation in the critical regions, while the compared methods generate a negative effect on the background. In image3 and image4, TCRN makes a flexible enhancement to the images to best match human visual perception. Image5 and image7, respectively, correspond to the complex structure and the dark scene image. Our method achieves satisfactory results without damaging the structure, while Water-net and FUnIE-GAN overcompensate and introduce red artifacts, respectively. Image6 is a demanding test for the color recovery ability of models. TCRN perfectly restores the color of the clownfish, but other methods can only remove the color cast.

In Figure 7, the degraded image that was extracted from the Test-60 dataset suffered from a thick haze effect. Our results eliminate the haze effect on image quality in subjective vision. We labeled the predicted images with EMBM for reference (the larger, the better).

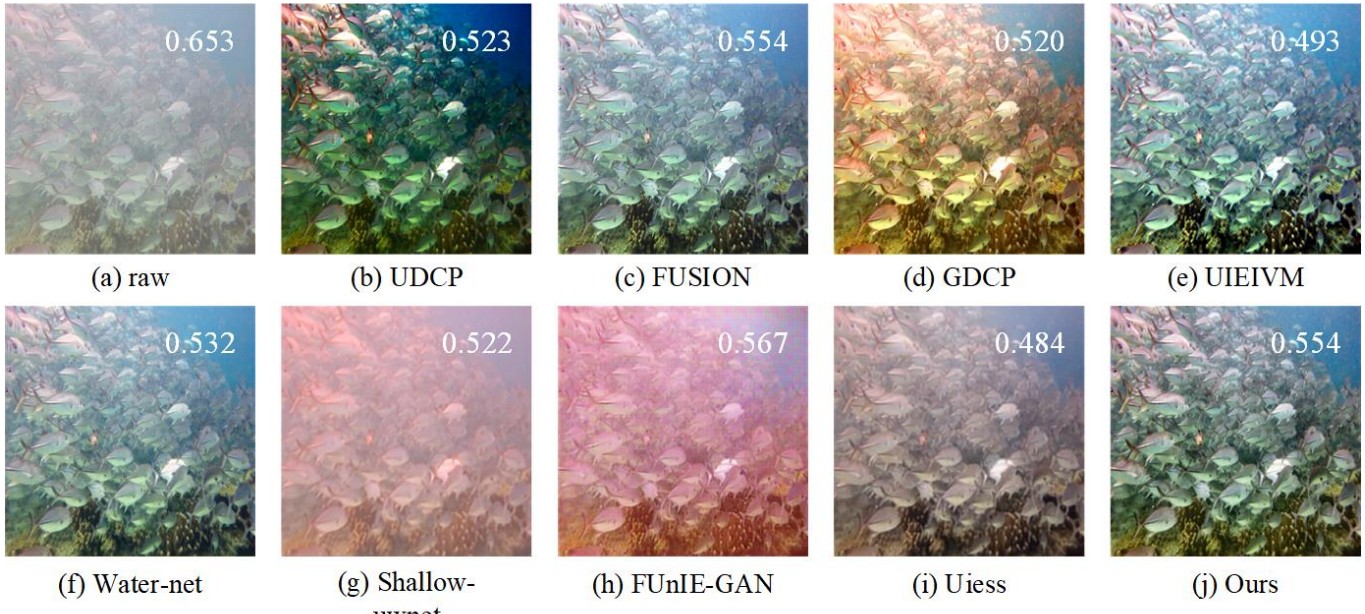

**Figure 7.** Visual comparisons on a typical real underwater image with obvious haze.

### 4.6. Ablation

In this section, we conduct sufficient ablation experiments to analyze the core components of TCRN, including the triple-color space feature extraction network (TTCN) and loss function, and the module ablation (MA) for a dense pixel-attention module (DPM), pre-fusion module (PFM), and group structure (GS). More specifically:

1. The designations w/o LAB, w/o HSV, and w/o HSV + LAB represent TCRN without LAB, HSV, and LAB+HSV color space feature, respectively.
2. The designations w/o DPM, w/o PFM, and w/o GS indicate a TCRN without a dense pixel attention module, pre-fusion module, and group structure.
3. w/o per loss indicates that only the $l_1$ loss function is included.

The quantitative full reference scores identified on Test-90 and Test-120 can be seen in Table 5; the contribution of the different color spaces to visual perception, the visual comparison of various modules, and the impact of perceptual loss on the results are shown in Figure 8. The conclusions drawn from the ablation study can be listed as follows:

**Table 5.** Quantitative results of the ablation study, in terms of average PSNR and SSIM values.

| Modules | Baselines | Test-90 | | Test-120 | |
|---|---|---|---|---|---|
| | | PSNR | SSIM | PSNR | SSIM |
| | Full model | 22.13 | 0.91 | 26.75 | 0.84 |
| TTCN | w/o LAB | 21.75 | 0.89 | 17.80 | 0.74 |
| | w/o HSV | 21.66 | 0.89 | 26.31 | 0.83 |
| | w/o LAB HSV | 21.29 | 0.88 | 26.66 | 0.83 |
| MA | w/o DPM | 21.07 | 0.89 | 26.65 | 0.83 |
| | w/o PFM | 21.92 | 0.89 | 26.63 | 0.83 |
| | w/o GS | 22.01 | 0.90 | 26.63 | 0.83 |
| LOSS | w/o per loss | 21.76 | 0.90 | 26.53 | 0.83 |

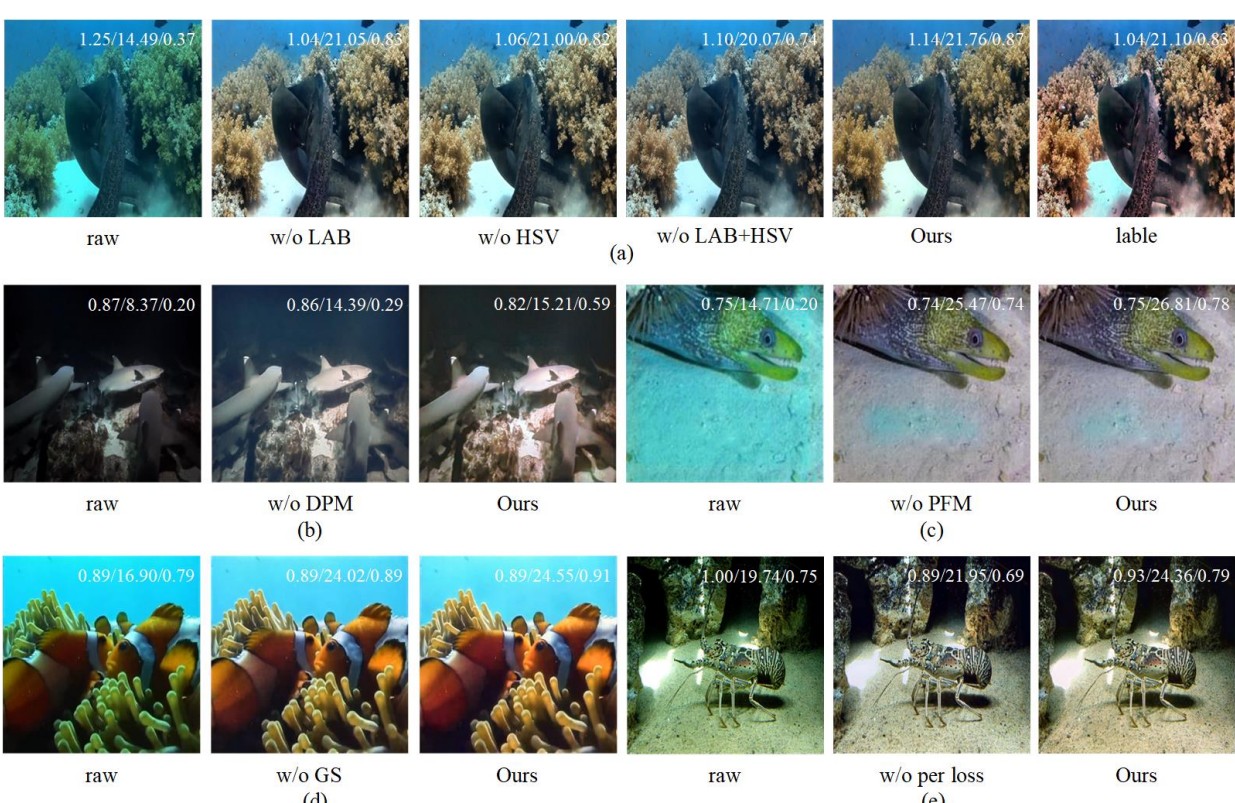

**Figure 8.** Visual comparisons with the ablation study. Here, (**a**) represents the TTCN examples, while (**b**–**e**) presents the MA and LOSS examples. The numbers on the top-right corner of each image refer to the PCQI/PSNR/SSIM (the larger, the better).

(1) The quantitative PSNR/SSIM scores on Test-90 and Test-120 can be seen in Table 5. The full TCRN model realized the best performance, which confirms the effectiveness of the core components.

(2) The contribution of different color spaces to enhancement is shown in Figure 8a and Table 5. Cutting out any color space will result in incomplete color correction. In contrast, TCRN shows better performance and proves the importance of triple-color space features for image reconstruction.

(3) For module ablation, the haze result generated by the w/o DPM can be seen in Figure 8b, which effectively verifies the performance of the DPM module for feature filtering in STEP I. In Figure 8c, the w/o PFM preserved a larger area of blue artifact than the full model; we speculate that the reason for this is due to the lack of feature interaction between the different color spaces, which leads to the model's inability

to accurately identify information. PFM successfully guided the interaction of different visual characteristics. As seen in Figure 8d, GS had a positive effect on color reconstruction.

(4)  As shown in Table 5, the influence of w/o per loss on quantitative metrics was weak. However, we can see from Figure 8e that the addition of perceptual loss leads to a better saturation and hue of the enhanced image.

Finally, we performed a set of ablation analyses on the selection of the number of spatial attention modules (SPBs). To ensure that the raw image is enhanced to a pleasing result in terms of hue, saturation, and brightness, the color quality evaluation metrics CCF [44] and MCNI [45] were employed to evaluate the number of SPBs on Test-515. The values of CCF and MCNI that correspond to the different numbers of SPBs are shown in Figure 9. We can see that CCF and MCNI reach the highest points of their curves when the number of SPBs equals 8. Therefore, 8 SPBs were selected to be the backbone of the group structure.

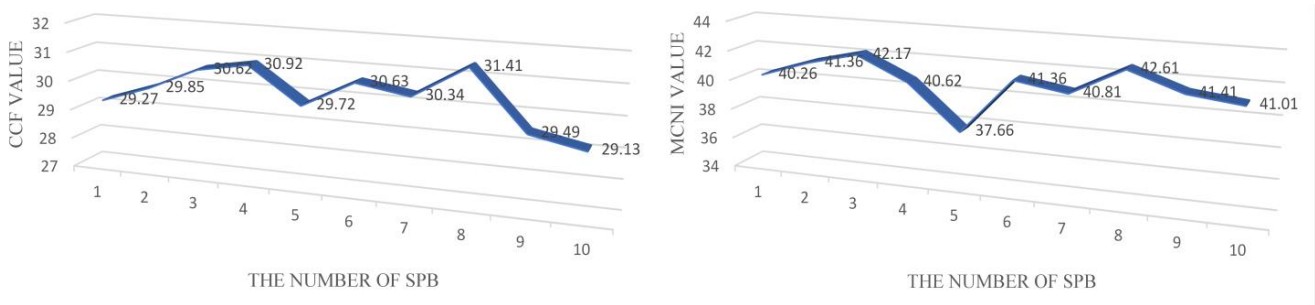

**Figure 9.** Ablation study of the contributions of SPBs, showing the CCF and MCNI comparison of different numbers of SPBs using the Test-515 dataset.

*4.7. Complexity Analysis*

In our study, Test-90 was used for the purposes of complexity analysis. As shown in Table 6, we computed the parameters for all deep learning methods and measured their corresponding time consumption values. Despite the impressive performance achieved by our method, there are still some limitations. We refrained from compressing the parameters in order to fully leverage the performance capabilities of both U-net and DPM, and the speed of TCRN was still guaranteed. Our research provides a novel idea for underwater image enhancement. In future research, we will design a more lightweight network.

**Table 6.** Model compression performance metric.

|  | **Water-Net** | **Shallow-uwnet** | **FUnIE-GAN** | **Uiess** | **Ours** |
|---|---|---|---|---|---|
| Parameters (m) | 1.23 | 0.24 | 7.73 | 2.23 | 22.37 |
| Per image (s) | 0.52 | 0.11 | 0.48 | 0.62 | 0.51 |

## 5. Conclusions

In this paper, we propose a two-step underwater image enhancement network aimed at improving the visual quality of the images acquired by marine autonomous detectors. In STEP I, the triple-color space feature extraction network successfully utilizes the parallel U-net-like structure employing the feature extraction module to extract different visual characteristic information types, such as hue, saturation, and brightness, and combines the information with the dense pixel attention module to ensure the presence of primary features while eliminating the haze effect due to interference. Moreover, the introduction of the pre-fusion module enables TCRN to comprehensively take into account various forms of visual feature information. In STEP II, the extracted features are compensated for by the fully connected layer, which promotes the dependency of high-dimensional features in a

channel-wise manner. Then, the group structure is used to reconstruct the image. Finally, our core components are proven to offer excellent performance by means of extensive experiments. Both qualitative and quantitative experiments have verified that our TCRN can complete image enhancement tasks in complex underwater scenes. In the future, we will continue to research this challenging underwater environment.

**Author Contributions:** Conceptualization, R.Z. and S.L.; methodology, R.Z.; software, R.Z.; validation, R.Z., Z.N. and J.L.; formal analysis, R.Z.; investigation, R.Z.; resources, R.Z.; data curation, Z.N.; writing—original draft preparation, R.Z.; writing—review and editing, R.Z.; visualization, Z.N. and J.L.; supervision, S.L.; project administration, S.L.; funding acquisition, S.L. All authors have read and agreed to the published version of the manuscript.

**Funding:** This research was supported by the National Key Research and Development Program of China (Grant No. 2018YFB1403303), and the Basic Scientific Research Project of Higher Education Institutions of Liaoning Provincial Department of Education (Grant No. LJKMZ20220615).

**Institutional Review Board Statement:** Not applicable.

**Informed Consent Statement:** Not applicable.

**Data Availability Statement:** Not applicable.

**Conflicts of Interest:** The authors declare no conflict of interest.

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
