# Peer review of "TCRN: A Two-Step Underwater Image Enhancement Network Based on Triple-Color Space Feature Reconstruction"

_jmse, doi:10.3390/jmse11061221_

Round 1
Reviewer 1 Report
This paper deals with underwater color enhancement. Results seems interesting but a lot of thing should be more accurately describded.
Line 62 : Why using simulatneously RGB, LAB, HSV. They contains the same information in different space. If those color spaces are really important, the network could learn to create them, don't you think ? Why not using other colorspaces ?
Line 65 : It is maybe a bit strange to talk about results in Figure 1 while none of the methods have been describded. Also, images are a bit small to see clear visual quality
In equation (4) why using L1 instead of L2 ?
In equation (6) why using 0.5 for lambda, did you try different values ?
Line 205 : I didn't find any Dark or UImagine in [30]. Only EUVP dataset is mentionned. You should spend more time presenting the different dataset, this is a critical point in deeplearning. I don't think you really have access to corrected, not corrected images for learning, hence this is the main drawback of the paper.
In section 4 you should compare to a very simple method like "histogram streching" which is very robust for its simplicity. For example, I tried on Figure 4 images and it gives very good visual effect, it worth knowing it for the reader.
line 297 : You cannot write "real color" you don't know what the real color was.
I don't find the ablation study very interesting, I know that a lot of people ask for that in deep but it is not always meaningful. You can keep it, just a remark.
Will the code be made opensource ? Otherwise the scope of this work will be very limited and not repeatable.
English is not so bad but need some rephrasing.
Reviewer 2 Report
Authors are presenting an U-net based deep neural network for underwater image enhancement. Proposed reconstruction network (TCRN) shows very good performance in the series of quantitative and qualitative tests.
However, there are some objections to the paper:
Authors should present proof or at least reason for introducing hypothesis that triple color input to the DNN should provide better results.
Since pixel values presented in LAB, HSV or RGB color spaces can be translated from one space to another with quite simple transformations, explanation why features from separated inputs are better than using only one input and some additional network layer should be given.
Statement in Introduction, lines 55 and 56 (deep learning approaches are divided to CNNs and GANs) may be misleading and should be corrected (in this form it sounds wrong).
In the list of contributions, third one (line 73) should be omitted.
There are also some minor inaccuracies in other parts of the paper (such as: Uimage in Table 1 or Uimagine in the text?).
Language needs significant improvements. Abstract is perhaps the worst part but numerous errors occur throughout the whole text (complete sentences should be rewritten but there are also "information lost", "sebsection" etc. errors).
Reviewer 3 Report
This manuscript presents a novel technique to enhance underwater imagery based on a deep learning methodology. The results show a significant improvement with respect to state-of-the-art methods. The manuscript is well-written, it has an appropriate organization and flow of ideas, the experiment design is also adequate to demonstrate technical feasibility, and the evaluation metrics are standard for image quality assessment.
Therefore, I recommend its publication in its present form.
Reviewer 4 Report
A two-step underwater image enhancement network is presented based on triple-color space feature reconstruction and the idea is clear. However, there exist some issues to be addressed.
1. The abstract can be elaborated. It will be better to include statements indicating the performance improvement of proposed method. For instance, improvement in accuracy of .... % for increase or decrease in computational complexity with factor of ..... times.
2. In introduction, more deep learning based methods can be introduced.
3. Please indicate limitations of the proposed method along with proper justification to be considered as a better method.
4. What about the computational complexity of proposed method in comparison to others ? please give some analysis.
It will be better to revise the manuscript to reduce the ambiguity in the writing. For instance,
i) The sentence in page 6 " ... for our method. As shown in Table 1." should have proper connection.
ii) Make sure that the usage of articles (a, an, and the) are correct throughout the manuscript.
Round 2
Reviewer 2 Report
Most of my comments were properly addressed. Language is improved although there are some minor needed corrections.
As written in previous section, some additional language corrections and polishing is needed.